# q2d: Turning Questions into Dialogs to Teach Models How to Search

**Yonatan Bitton**[†,‡,*]    **Shlomi Cohen-Ganor**[‡]    **Ido Hakimi**[‡]
**Yoad Lewenberg**[‡]   **Roee Aharoni**[‡]   **Enav Weinreb**[‡]
[†] The Hebrew University of Jerusalem; [‡] Google Research
{yonatanbitton,shlomic,idohakimi,yoadlew,roeeaharoni,eweinreb}@google.com

## Abstract

One of the exciting capabilities of recent language models for dialog is their ability to independently search for relevant information to ground a given dialog response. However, obtaining training data to teach models how to issue search queries is time and resource consuming. In this work, we propose *q2d*: an automatic data generation pipeline that generates information-seeking dialogs from questions. We prompt a large language model (PaLM) to create conversational versions of question answering datasets, and use it to improve query generation models that communicate with external search APIs to ground dialog responses. Unlike previous approaches which relied on human written dialogs with search queries, our method allows to automatically generate query-based grounded dialogs with better control and scale. Our experiments demonstrate that: (1) For query generation on the QReCC dataset, models trained on our synthetically-generated data achieve 90%–97% of the performance of models trained on the human-generated data; (2) We can successfully generate data for training dialog models in new domains without any existing dialog data as demonstrated on the multi-hop MuSiQue and Bamboogle QA datasets. (3) We perform a thorough analysis of the generated dialogs showing that humans find them of high quality and struggle to distinguish them from human-written dialogs. [1]

## 1 Introduction

Recent dialog generation models, such as LaMDA (Thoppilan et al., 2022), BlenderBot3 (Shuster et al., 2022b) and Sparrow (Glaese et al., 2022) use an external search API to generate grounded and factually accurate responses (Parisi et al., 2022). This is important for providing reliable and consistent answers (Shuster et al., 2022a), especially

when discussing entities and asking related questions with anaphora. To do this, these models use a query generation component that is trained on dialog-to-search-query datasets. When the model is triggered with a dialog turn that requires search, it generates a query that is used to obtain a search result, which is then used to generate a grounded response. This allows the model to provide relevant information about the world in its responses to user queries. For example, a model trained in 2021 should be able to provide a factual response to the question "How old is Joe Biden?" even in 2023. In a conversation, one might discuss an entity (e.g. "Joe Biden") and later ask a related question (e.g. "How old is he?") with anaphora. In order to provide reliable and consistent answers, it is necessary to generate a decontextualized query (e.g., "How old is Joe Biden") for a search engine.

Using APIs also decouples language and reasoning from knowledge (Borgeaud et al., 2021; Parisi et al., 2022), which can help prevent errors caused by outdated information being stored in the model's parameters. For example, if a model trained at the end of 2021 is asked "How old is the current president?", it may produce the incorrect query "How old is Donald Trump" if its parameters are outdated or if it provides factually-inconsistent responses (a.k.a "hallucinations").

Query generation datasets have been created using human annotators, limiting them in scale, control, and quality (Komeili et al., 2021). As a result, when a new domain is introduced, a significant amount of human effort is required to create a new query generation dataset for that domain (Gupta et al., 2021; Dziri et al., 2021). The fact that language models often generate hallucinations (Zhao et al., 2020; Maynez et al., 2020; Lee et al., 2018), especially in new domains or dialogs that differ from the training data (Nie et al., 2020; Honovich et al., 2021, 2022a), highlights the need for more

---

[*]Work done during an internship at Google Research.

[1]We publicly release all prompts and will release all generated datasets.

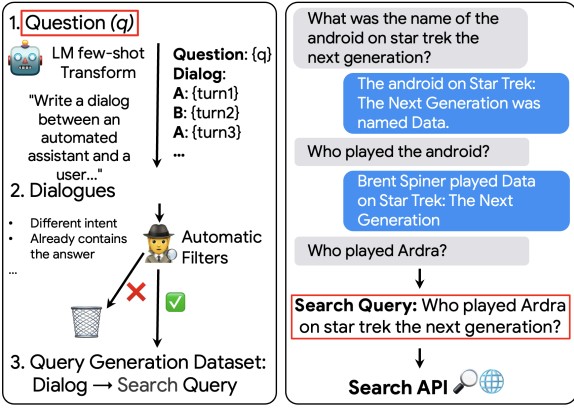

Figure 1: Left: Our *q2d* method starts from an existing query or question and prompt a few-shot language model to transform it into a dialog. We filter out cases where the intent of the generated dialogue differs from the intent of the initial query and apply additional filters. Right: We take a question from the QReCC dataset (surrounded by a rectangle) and generate an information-seeking dialog with *q2d*. By starting with a query and generating a dialog, we create {dialogue → query} dataset, which is used to train and evaluate query generation models, which communicate with an external search API to generate factual responses.

effective query generation datasets that will foster more grounded and factually consistent models.

In this work, we propose a data generation pipeline to improve grounded dialog models with access to search engines. To create a dialog-to-search-queries dataset for training the query generation component in such models, we reverse the process, starting from a search query and generating an information-seeking dialog that corresponds to that query. Our automatic pipeline, shown in Figure 1, begins with a search query or question, and prompts a large language model (PaLM; Chowdhery et al., 2022) to generate a conversational dialog that conveys the information need implied by the given query. For example in Figure 1, we take the question "Who played Ardra on star trek the next generation?" from the Natural Questions dataset (Kwiatkowski et al., 2019) and generate a dialog with a similar intent: the correct answer to the original question ("Marta DuBois") is also a correct response to the generated dialog. This process allows us to leverage existing question-answering datasets, which are widely available for different domains, and extend them by generating dialogs that preserve the original information need while controlling the dialog domain and style.

To assess whether the automatically generated dialogs can replace human-generated dialogs, we experiment with QReCC NQ (Anantha et al., 2020), a human-curated dialog dataset. We generate a training set that is the same size as the original dataset, but with synthetic dialogue, and use it to train a query generation model. The resulting model obtains 90%–95% of the performance of models trained on the human-generated training data, using the same metrics used to evaluate QReCC (Anantha et al., 2020).

Other than training query generation models, our approach is also useful for training the dialog generation models themselves when no dialog data is available for a new domain. We demonstrate that on the domain of multi-hop question answering, where we first show that existing dialog models struggle to perform well on a domain-specific challenge set. We then generate synthetic dialog data from the MuSiQue (Trivedi et al., 2021) multi-hop QA dataset, and show that training a dialog model on this data improves performance.

We provide a thorough analysis of the quality of the generated datasets, demonstrating that they (a) looks natural, and humans struggle to distinguish the synthetic dialogs from natural; (b) factual: generated and human-annotated answers perform similarly in query generation; (c) correct: dataset labels are accurate, and strict filtering improves results.

To conclude, our main contributions are:

1. We introduce *q2d*: an automatic method to generate information-seeking dialogs from questions using large language models.

2. We show that our method is beneficial for training query generation and dialog generation, including in different domains like multi-hop QA.

3. A thorough analysis showing that the synthetically generated dialogs are natural, factual and correct.

4. Publicly releasing the generated datasets and generation prompts, code, and evaluation protocols.

## 2 Generating Dialogs from Questions

In this section, we describe our automatic method, called *q2d*, for generating dialogs from questions, and the properties of datasets produced by this

**Algorithm 1** Generate Dialogues from Questions

**input**
    Few-Shot Model $M_{fs}$, QA Dataset $(Q, A)$,
    Examples Queries $S_q = \{(q_i, d_i)\}_{i=1}^{k}$,
    Examples Dialogues $S_d = \{(d_i, q_i)\}_{i=1}^{k}$,
    Instructions Query $I$, Instructions Dialogue $I_r$,

**execute**
    $dataset \leftarrow \emptyset$
    **for** $(q, a) \in (Q, A)$ **do**
        $dialogue \leftarrow M(S_q, I, q)$
        $q' \leftarrow M_{fs}(S_d, I_r, dialogue)$
        **if** filter(dialogue,q,q',a) **then**
            $dataset.add((dialogue, q, a))$

**output**
    Query Generation Dataset: $D = \{(d_i, q_i)\}_{i=1}^{|Q|}$

method. Our goal is to reduce the effort associated with generating a training dataset for training generation, and to improve query-generation-based dialog models with a high-quality training dataset. Query generation can start by extracting queries from existing dialogs. However, our approach is unique in that it begins with factual queries or questions, allowing us to leverage existing resources. Any question-answering dataset, queries dataset, or queries used in popular web search services or dialog model logs can be used with our algorithm.

The algorithm is described in Algorithm 1 and consists of three main steps:

1. Starting from a query or question from the set $Q$, we use a few-shot model $M_{fs}$, specifically we use PaLM, and instructions $I$ to generate a dialog given the query. The few-shot prompts can be manually written to adapt to different conversation styles, or sampled from existing dialogs dataset.

2. Using the same few-shot examples in reverse, $S_d$ and $I_r$, we generate a query based on the generated dialog, $q'$.

3. Filtering: we filter dialogs with different intent, or dialogs where the dialog answer is contained in the dialog. We elaborate on the different filters below.

**Filtering.** In this part we attempt to filter (dialog, query) samples that would not be beneficial for training or testing. We do it in three steps, elaborated below. We stress that there are many more filtering strategies possible, and exploring

them is left for future work. First, we filter out dialogs whose intent is different from the original query by measuring the similarity between the query and its reversed version using SBERT similarity ($sim(q, q')$) and comparing it to a threshold ($T_{query}$). If the similarity is below the threshold, the generated query is considered to have a different intent and the dialog is filtered. Appendix A, Section A.2 shows several examples of dialogs, original and reversed query and SBERT semantic similarity. Second, we filter out cases where the answer is included in the dialog by measuring the n-gram overlap between the dialog and the answer using the Rouge metric (Lin, 2004). If the overlap is above a threshold ($T_{answer}$), the answer is entailed in the dialog and the example is filtered. For example, if the final answer ("Marta DeBois") would have been already written in the dialog for the role of playing *Ardra*, the final question ("Who played Ardra") would not make sense. Finally, we filter out cases where the last turn of the dialog is similar (>80%) to the original question using SBERT similarity. These cases include situations where no anaphora is required.

In this work, we use PaLM (Chowdhery et al., 2022), a large language model with 540B parameters, as the few-shot language model for generating dialogs with a temperature of $0.6$. We provide a fully working code with GPT-3 (Brown et al., 2020) for reproducibility. The set of prompts and instructions can be found in Appendix A, Section A.3. For the similarity metric ($sim$), we use the *all-mpnet-base-v2* model from Sentence Transformers, with a threshold similarity of $T_{query} = 0.999$. This threshold is justified through human-evaluation and ablation studies for the filtering in Section 5.3.

## 3 Replacing Human-Annotated with Auto-Generated Data

In this section, we evaluate the extent to which our automatically generated dataset can replace the human-annotated dataset. We use the QReCC NQ dataset (Anantha et al., 2020), which contains (dialog, query) pairs, and automatically generate a dialog from natural questions. This allows us to create an automatically generated train set of the same size, and compare it to the human-annotated dataset. An example of a human-generated dialog compared to an automatically generated dialog is shown in Figure 2. We use the version of the dataset where the intermediate questions are con-

**Q: Was the film The Vikings based on a novel?**

Human Annotated → Auto-Generated

*Human Annotated:*
- Where is the castle in The Vikings film?
  - Aella's castle in the film, The Vikings, was the real Fort de la Latte in north east Brittany in France
- Who directed the film?
  - The Vikings was directed by Richard Fleischer
- Was the film based on a novel?

*Auto-Generated:*
- what year was the film The Vikings released?
  - The film The Vikings was released in 1958.
- Who directed the film?
  - Richard Fleischer directed the film.
- Was it based on a novel?

Figure 2: An example of human annotated dialogue from QReCC and an automatically generated dialogue produced for the same question.

| Model | Training Dataset | SBERT Similarity | Rouge-1 Recall | Search Results Recall@10 |
|---|---|---|---|---|
| T5 | Human Annotated | 92.4 | 88.1 | 68.5 |
|  | Auto Generated | 87.5 (**95%**) | 83.3 (**95%**) | 61.5 (**90%**) |

Table 1: Results on the human-annotated QReCC NQ test set, experimenting with replacing the human-annotated data with automatically generated data with the *q2d* method. Bold shows the percentage of performance for a model trained with auto-generated data out of a model trained with human-annotated data. Training on the automatically generated data achieves 90%-95% of the model trained on the human annotated results.

textualized, rather than decontextualized. For example, the second and third user turns in the figure are contextualized versions of the decontextualized questions "Who directed the film, The Vikings?" and "Was the film The Vikings based on a novel?".

**Dataset Generation.** To generate our dataset, we use our *q2d* method as described in Section 2. For the few-shot examples of queries and dialogs ($S_q$ and $S_d$), we sample 15 examples from QReCC that fit within the maximum input sequence length. These examples are available in Appendix A, Section A.3. For the base questions ($Q$), we use the Natural Questions (Kwiatkowski et al., 2019) dataset instead of the QReCC NQ questions to reduce dependence on QReCC. Importantly, all of the questions and dialogs in the natural and automatically generated datasets are disjoint. In total, we generate 13K samples, the same as the QReCC NQ train set. Full prompts, instructions and examples are available in Appendix A, Section A.1.

**Metrics and Models.** Our metrics are the same as those used in the QReCC dataset, comparing the original and generated queries. These include Rouge-1 Recall (Lin, 2004) for measuring the similarity between two text unigrams, and SBERT embedding semantic similarity for comparing the semantic content of two sentences (same metric as in §2).[2] We also use Recall@10 to compare the retrieved URLs for the ground-truth query and the generated query.[3] We conduct experiments using

an open-source T5-3B model (Raffel et al., 2020) in its original form (referred to as 'None'), by fine-tuning it on the natural QReCC training data and contrasting the results with those obtained from training on the auto-generated QReCC dataset. We use a batch size of 32, an Adam optimizer, a learning rate of 0.0001, and fine-tune it for 10,000 steps.

**Results.** Results are presented in Table 1. We observe that by replacing human annotated data with auto generated data we were able to reach 90%–95% of the results with a set of the same size using the same model, demonstrating the efficacy of our *q2d* approach in minimizing annotation labor and producing synthetic training data that is nearly as effective as human-annotated data.

## 4 Extending Query Generation: Multi-Hop QA

This section shows that our method is effective as a benchmark and training signal that generalizes to human-annotated data. It is also flexible and able to adapt and improve for specific styles of dialog, even without annotated data. It allows us to create dialogs similar to a target domain and provide a fully labeled query-generation dataset. The generated data is useful for training and evaluation, as well as exploring model performance in new scenarios. We demonstrate this using a multi-hop question answering example.

**Manual Dialog Construction.** We define a challenging test set for multi-hop dialogs by annotating the Bamboogle dataset (Press et al., 2022), which consists of 125 multi-hop human-constructed questions. We create dialogs that ask the same questions, with the user as the information seeker and the assistant as the information provider. The assistant should help the user obtain the information they are seeking, clarify any questions, and move

---

[2]We replaced USE (Cer et al., 2018) with SBERT MPNet embeddings which are perform better on the STS benchmark (Cer et al., 2017) (75 → 88).

[3]https://serpapi.com/ provides an open API for a popular internet search engine.

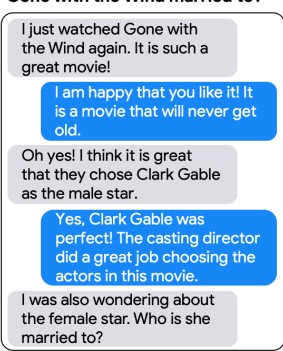

**Q: Who is the female star in Gone with the Wind married to?**

> I just watched Gone with the Wind again. It is such a great movie!

>> I am happy that you like it! It is a movie that will never get old.

> Oh yes! I think it is great that they chose Clark Gable as the male star.

>> Yes, Clark Gable was perfect! The casting director did a great job choosing the actors in this movie.

> I was also wondering about the female star. Who is she married to?

Figure 3: An example dialog generated by *q2d* from a MuSiQue multi-hop question. The dialog's final query has a 0.5034 semantic similarity score with the original, indicating different semantic content and thereby not being filtered.

the conversation forward without trying to mimic human-to-human interaction. Figure 3 shows a positive example where the generated dialog is not being filtered. On the other hand, a negative example could be a query like "Who is the *female* star in Gone with the Wind married to?" which closely resembles another query asking about the *male* star, with a high similarity score of 0.9595. This demonstrates the method's shortcomings in filtering semantically similar queries.

Full instructions, examples and annotated data can be found in the Appendix A, Section A.4, including examples with model predictions.

**Dataset Generation.** We use our *q2d* method as described in Section 2 to generate dialogs that ask multi-hop questions, using the MuSiQue dataset (Trivedi et al., 2021) as the base for the questions ($Q$). MuSiQue is a challenging multi-hop QA dataset that is partially auto-generated, so we generate dialogs from partially generated questions. This illustrates how we can use automatically generated data to improve on human-annotated data. We use seven few-shot examples ($S_q$ and $S_d$). As a result, we generate 3K train samples and 480 test samples. Full prompts, instructions and examples are available in Appendix A, Section A.1.

**Metrics.** The metrics used in this work are the same as those described in the previous section: Rouge-1 Recall, SBERT embedding semantic similarity, and Recall@10.

**Models.** We evaluate several state-of-the-art language and dialog models. These include PaLM 540B (Chowdhery et al., 2022), Flan-U-PaLM 540B (Chung et al., 2022), T5-3B (Raffel et al., 2020), BlenderBot3-3B (Shuster et al., 2022b), WizInt Search Engine FiD (Lewis et al., 2019). These models are used in a zero-shot setting, except for T5, which is fine-tuned on the auto-generated

| Model / Test Set | SBERT Similarity | | Rouge-1 Recall | | Search Results Recall@10 | |
|---|---|---|---|---|---|---|
| | M | B | M | B | M | B |
| WizInt | 66 | 67 | 40 | 36 | 21 | 21 |
| BlenderBot3 | 62 | 69 | 32 | 35 | 19 | 24 |
| T5 (QReCC) | 74 | 77 | 70 | 65 | 34 | 37 |
| PaLM 540B | 88 | 82 | 81 | 69 | 52 | 41 |
| Flan-U-PaLM 540B | 89 | 82 | 83 | 68 | 57 | 39 |
| **T5 (MuSiQue)** | 97 | 91 | 94 | 80 | 75 | 54 |

Table 2: Performance of language and dialogue models on query generation test sets is shown. 'M'`B' indicates results on MuSiQue auto-generated Bamboogle manually constructed dialogues. (QReCC) and (MuSiQue) indicate fine-tuning on a "q2d" dataset. Best results were achieved by models fine-tuned on MuSiQue auto-generated dialogue, which improved T5 results by 14%-59% on the human-annotated test.

MuSiQue dialogs in the same method presented in Section 3. BlenderBot3 and WizInt are publicly available in Parlai (Miller et al., 2017), exact details and versions are described in Appendix A, Section A.7. More details on the instructions for zero-shot models can be found in the Appendix A, Section A.3.

**Results.** Query generation results are presented in Table 2.[4] Qualitative examples with T5 model predictions are available in Appendix A, Section A.1. The T5 model improves performance on the human-curated Bamboogle test by 14%-59% after fine-tuning on the auto-generated MuSiQue multi-hop dialogues. We show examples for it in Appendix A, Section A.6. This improvement also correlates with improvements on the auto-generated test set, indicating the effectiveness of our method for creating evaluation data. To conclude, our results show that our datasets are effective as a benchmark for query generation, as well as training data that generalizes to both auto-generated and human-annotated test sets.

**Producing a Partially Decomposed Query.** Given a multi-hop dialog, query generation models may resolve partial information. For example, if a dialog asks "How old is the current US president?", a query generation model may produce "How old is Joe Biden?", which is correct at the time but may become outdated in the future, or may produce hallucinations. To prevent this, we can make

---

[4]We show relatively low scores with WizInt and BlenderBot3 that seem to be oriented in finding the topic query rather than concrete questions.

two query generation calls (first to discover the current US president and then their age), decouple knowledge from executing (Borgeaud et al., 2021; Parisi et al., 2022), periodically update the model's weights, or disallow the model from making partial resolves. This will help ensure that the generated query remains accurate and relevant over time. The fine-tuning technique described in this section uses the last approach to avoid making assumptions about the current president's age or identity.

# 5 Intrinsic Evaluation: Naturalness, Factuality and Correctness

In this section we perform a thorough analysis of the generated dialogs, focusing on the QReCC NQ dataset which contains human annotated dialogs, and evaluate their naturalness (§5.1), factuality (§5.2) and correctness (§5.3).

## 5.1 Naturalness: Humans Struggle to Distinguish Synthetic Dialogs from Natural

We define a human-evaluation task to distinguish between naturally generated dialogs and auto-generated dialogs. We sample 100 annotated dialogs from QReCC NQ (Anantha et al., 2020) and mix them with 100 dialogs we generated. The annotators, who are not the authors of the paper and have a STEM degree, were asked to mark 1 if the dialog seems to be generated by a machine, and 0 otherwise.[5] The labels were hidden. We use three annotators for each sample and select their majority vote as the final answer. The results show that the majority vote achieved a success rate of 50.5%, while the random chance is 50%. All individual annotators achieved between 50%–55% in this task. In 26% of the cases there is a full agreement between all three annotators. When all agreed, the result improves to 51.9%, which is still close to random chance. These results indicate that humans struggle to differentiate between natural and auto-generated dialogs. This suggests that the auto-generated dialogs are of high quality and are similar to human annotations, and can be used in place of human-generated dialogs in certain situations, saving time and resources.

---

[5]Full instructions to the annotators are provided in Appendix A, Section A.5.

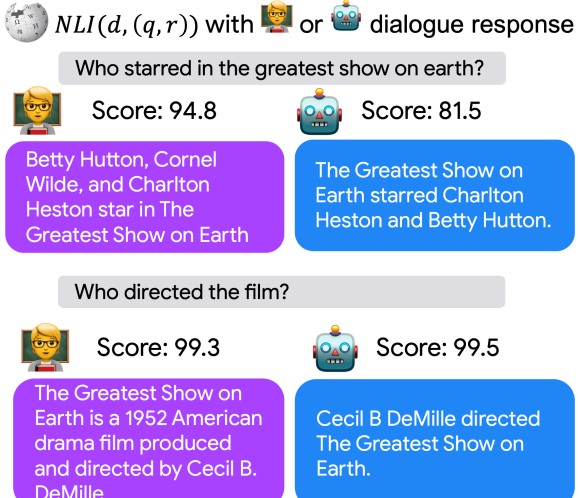

Figure 4: Illustration of the response factuality evaluation. For each turn, we produce a response with PaLM, and compare the generated response to the human annotated response. We use an NLI model to score whether the response answers the question ("Hypothesis: The answer to the question {q} is {r}") according to the Wikipedia document $d$ used by the human annotator in the ground-truth response generation ("Premise: {d}"). In the first response there is a lower score for the PaLM response because it misses the mention of 'Cornel Wilde' that appears in the document summary.

## 5.2 Factuality: Generated and Human-Annotated Answers Perform Similarly in Query Generation

The q2d method generates a dialog by starting with a query and generating a series of related questions and answers. However, since the intermediate answers are generated by a large language model, there is a chance that they may be factually correct or incorrect. This raises the following questions. (1) Are the intermediate answers factually correct? (2) How does the factuality of the generated answers affect the results of downstream tasks?

We replace all human annotated answers in the QReCC NQ training split with PaLM generated answers. To produce PaLM answers, we use a few-shot prompt, where the input is the original dialog ending in a question, and the output is the PaLM response. An example is provided in Figure 4.

**Intermediate Answers Factuality According to Automatic Metrics and Human Raters.** To answer the first question, we evaluate the factual correctness of the generated answers by using an NLI (Dagan et al., 2005) model presented by Honovich

et al. (2021). We take the question ("q"), the response ("r") that may be the ground-truth annotated response or the generated response, and the Wikipedia document ("d") summary available in QReCC dataset. We construct the following NLI instance: "premise: {d} hypothesis: The answer to the question {q} is {r}' and produce NLI scores for the ground-truth responses vs. the generated responses. Figure 4 illustrates our process. The average NLI scores for the human responses are 62%, and for the PaLM responses is 38%. However, this measure is biased towards the human responses since we measure it with the Wikipedia document that was used to generate the answer. PaLM might also produce a correct answer, that is just not written in the same exact words in Wikipedia. To test this, we conducted an annotation task with an annotator that is not a part of the paper authors. The annotator was presented with a 50 samples of dialog, query, and two options: A and B. One of the options was the original answer and the other was the generated answer. The annotator's task was to mark 0/1 for each answer indicating whether it was factual and relevant for the question. The results are that PaLM responses were marked as correct in 82% of the cases, compared to 93% correctness of the human responses. This result indicates the factuality and relevancy of the generated responses.

**For Query Generation, Generated Answers Perform Similar to Human-Annotated.** To answer the second question, we replace all of the human annotated answers with automatically generated answers, receiving a semi-auto-generated training set with the same structure and same annotated questions, but with PaLM generated dialogs. Then we train a T5-3B (Raffel et al., 2020) model on the human annotated and the semi-auto-generated version and compare the results. For example in Figure 4, the semi-auto-generated dialog is the one with the answers on the right side. We train the same way as we presented in Section 3. The results are 86.6% Rouge-1 Recall with the semi auto-generated training set, only a small drop (1.5%) from the results of the model trained on the natural data, indicating that although PaLM sometimes (<48%) produce in-factual responses, it only has negligible effect on the query generation task.

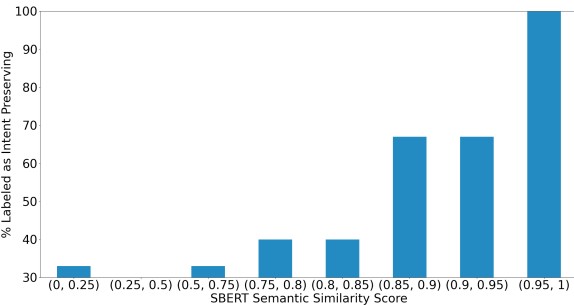

Figure 5: Intent-preserving annotation task results. The proportion of samples that were annotated as intent-preserving increases with the semantic similarity score.

### 5.3 Correctness: Generated Datasets Labels are Accurate, and Strict Filtering Improves Results

Our main filter measures a similarity between the original query and the reversed query $sim(q, q')$ and compare it to a threshold $T_{query}$. We measure its effect in human-evaluation and automatic ablation studies. Both experiments indicate the label correctness for the task of predicting the query from a dialog and the value of stricter filtering threshold.

**Humans Find that Dialogs Generated by Queries Have the Same Intent.** We define a human annotation task to determine whether the dialogs are intent-preserving. Annotators were asked to mark 1 if the dialog is intent-preserving, and 0 otherwise.[6] We use three annotators for each sample, and select their majority vote as the final answer. We follow the notation suggested by (Groenendijk and Stokhof, 1984) about entailment between questions: an interrogative $q$ entails another $d$ iff every proposition that answers $q$ answers $d$ as well (Jiang and de Marneffe, 2022). Here, $q$ stands for a question and $d$ stands for an information-seeking dialog. We defined eight SBERT semantic similarity score buckets, with 15 in each, covering all similarities between 0 and 100. Results are presented in Figure 5. All three annotators agree in 88% of the cases. The proportion of intent-preserving annotations grows according to the SBERT semantic similarity score, with a strong gain between 0.95 and 1, the only bucket with 100% intent-preserving annotations. Accordingly, we only select samples that have generated

---

[6]Full instructions to the annotators are provided in Appendix A, Section A.5.

| SBERT Similarity Filtering Threshold | Filtering Proportion | Query Generation Rouge-1 Recall |
|---|---|---|
| 0 | 0 | 68 |
| 0.25 | 6 | 69 |
| 0.5 | 16 | 72 |
| 0.75 | 37 | 74 |
| 0.8 | 44 | 76 |
| 0.9 | 62 | 79 |
| 0.95 | 72 | 81 |
| 0.99 | 84 | 83 |
| **0.999** | **88** | **84** |

Table 3: Reversed queries similarity filter. The similarity is measured between the original query $q$ and the reversed query $q'$ predicted with the few-shot model $q' \leftarrow M_{fs}(S_d, I_r, dialogue)$. The higher the filter threshold (strict filter), the better the results.

queries very similar to the original query ($\geq 0.99$) in the filtering step.

**Strict Filtering Leads to Higher Quality Data, Resulting in Improved Downstream Results.**
We measure different thresholds tested on an evaluation set of 1,000 instances we generated from other train queries. We also add another filtering method based on an NLI (Dagan et al., 2005) model, given a dialog "d" and a question "q", we construct the following NLI sample: "premise: {d} hypothesis: The dialog asks the question {q}", with different thresholds. Results are presented in Table 3. We report the Rouge-1 recall on the evaluation set. We see that performance increases as the reversed similarity threshold rises, and with a clear trade-off with the filtering proportion. The more data we generate, we are able to apply a more strict filtering, receiving higher quality data, that leads to better results.[7] We produced four options for the NLI-based method, with thresholds ranging from 0.65 to 0.82, and above it filtered too much data (below the goal of 13K). The max performance for the 0.82 threshold group is 70%, much lower than the alternative reverse queries filter.

## 6 Related Work

Our work relates to data generation, query generation for search-based models, and information retrieval datasets.

**Data Generation** Several works have used large language models for data generation (Agrawal et al., 2022; Honovich et al., 2022b; Yarom et al., 2023). Dai et al. (2022b) applies this technique to information retrieval, creating retrievers based on generated data that generate queries given the document. Their method involves round-consistency filtering using a large language model, a method similar to reverse translation. In the context of dialog generation, Dialog Inpaintint (Dai et al., 2022a) starts from a document and generates a dialog. Moreover, Gekhman et al. (2023a) introduced TrueTeacher, a synthetic data generation method that employs large language models for annotating model-generated summaries, demonstrating its effectiveness over existing techniques. Our approach focuses on generating dialogs from queries, which allows us to leverage the availability of existing QA datasets. This enables us to create information-seeking dialogs with the same intent as the original questions, along with automatically generated labels for the queries and answers.

**Search Based Query Generation** dialog models like LaMDA and BlenderBot use search APIs to generate factual responses. Training and evaluation data for such models is obtained mostly with human annotated data. Previous works (Shuster et al., 2022b; Thoppilan et al., 2022; Komeili et al., 2021) evaluated only the end-to-end dialog response without evaluating the generated query. The evaluation was primarily based on automated metrics of perplexity and F1, or with human annotations assessing whether the model response is sensible, specific, and interesting (SSI), or whether it is correct, engaging, and consistent. The evaluated dialogs were general, not necessarily information-seeking. The focus of this paper is on the query generation task for information-seeking dialogs, with a concrete question and an expected response.

**Question Rewrite** Works like QReCC (Anantha et al., 2020), Question Answering in Context (QuAC) (Choi et al., 2018), TREC Conversational Assistant Track (CAsT) (Dalton et al., 2020), QuAC and CANARD (Elgohary et al., 2019) in the information retrieval domain use human annotated data, that mostly contain follow-up dialogs, questions followed by answers. In the domain of Conversational Question Answering (CQA), a comprehensive study was conducted on the robustness of dialogue history representation, underscoring

---

[7]The high filtering proportion means that we simply need to generate more data (requiring more compute time) in order to achieve a dataset of the same size without filtering.

the significance of evaluations centered on robustness (Gekhman et al., 2023b). Our work focuses on the application of dialog models like LaMDA and BlenderBot, which often involve the use of less formal language and more human-like conversations. The need for a variety of query generation datasets has motivated us to develop an automatic method for generating dialogs for the query generation task, with a range of different styles and skills required.

# 7 Conclusions

We introduced *q2d*, a data generation pipeline that produces dialogs based on questions. We demonstrated that our method can replace human-annotated data to train query-generation models, and to create effective, natural, factual, and accurate evaluation and training data in new domains, even when no existing dialogue data is available.

# 8 Limitations

*q2d* comes with a set of limitations about costs, domain identification and factuality.

**Computational Costs.** The process of auto-generating data with large language models, although faster and more scalable than human annotations, still incurs significant computational costs. These costs, however, are expected to decrease as AI technologies advance.

**Domain Identification and Sample Selection.** Defining the target domain and selecting representative few-shot examples requires manual oversight. This step, although crucial for ensuring the diversity and representativeness of generated dialogs, adds a layer of complexity and time to the process.

**Factuality of Dialogs.** Our method generates dialogs that are generally factual but occasionally inaccurate. Although our analysis show these discrepancies do not impact query-generation tasks, they may challenge tasks where factuality is critical. Future applications should consider this limitation and potentially enhance factuality.

**Scope of Fine-Tuning and Model Improvements.** In this research, our primary aim was to show that larger foundation models' generated data, such as PaLM 540B, can significantly benefit more compact models like T5-3B. Specifically, when smaller models are fine-tuned on our auto-generated data, they can achieve performance surpassing the larger

foundation models in query-generation tasks. However, we did not explore the possible improvements to the original foundation models, like PaLM, when fine-tuned using our generated data. This represents a promising avenue for further research.

# 9 Ethics and Broader Impact

This paper is submitted in the wake of a tragic terrorist attack perpetrated by Hamas, which has left our nation profoundly devastated. On October 7, 2023, thousands of Palestinian terrorists infiltrated the Israeli border, launching a brutal assault on 22 Israeli villages. They methodically moved from home to home brutally torturing and murdering more than a thousand innocent lives, spanning from infants to the elderly. In addition to this horrifying loss of life, hundreds of civilians were abducted and taken to Gaza. The families of these abductees have been left in agonizing uncertainty, as no information, not even the status of their loved ones, has been disclosed by Hamas.

The heinous acts committed during this attack, which include acts such as shootings, sexual assaults, burnings, and beheadings, are beyond any justification.

In addition to the loss we suffered as a nation and as human beings due to this violence, many of us feel abandoned and betrayed by members of our research community who did not reach out and were even reluctant to publicly acknowledge the inhumanity and total immorality of these acts.

We fervently call for the immediate release of all those who have been taken hostage and urge the academic community to unite in condemnation of these unspeakable atrocities committed by Hamas, who claim to be acting in the name of the Palestinian people. We call all to join us in advocating for the prompt and safe return of the abductees, as we stand together in the pursuit of justice and peace.

This paper was finalized in the wake of these events, under great stress while we grieve and mourn. It may contain subtle errors.

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

# A  Appendix

## A.1  Generated Examples

Figure 6 and Figure 7 show *cherry picked* examples of human-annotated / auto-generated examples from Bamboogle / MuSiQue.

Tables 4 and Tables 5 show *random* examples from the auto-generated QReCC / MuSiQue dialogs.

## A.2  Filtering Examples

Table 6 show examples of generated samples with different SBERT similarity scores used for filtering.

## A.3  Prompts and Instructions

Prompts (instruction + few-shot examples) for the auto-generated dialogs are presented in Table 7 for MuSiQue, and Table 8 for QReCC NQ. We show three prompts for each dataset due page page length, full prompts files are here: MuSiQue, QReCC.

The reverse generation model uses the same examples but in reversed order (to generate a query given a dialog).

The instruction for the reverse generation model and the PaLM zero-shot is: "Given a dialog that asks an indirect question, extract the concrete question".

The instructions for the Flan-U-PaLM-540B and are a bit more detailed (we found out it worked better for this instruction fine-tuned model): "Your task is to reformulate the last conversational query in a dialog to a fully specified, context-independent query that can be effectively handled by Google Search.".

## A.4  Human dialog Generation for Bamboogle

Bamboogle human constructed dialogs are available here: Bamboogle dialogs.

Instructions for annotator: "Your task is to write a dialog given a question. You may use the internet in order to add factual information to dialog. The dialog intent should be understandable yet challenging. The responses should be 'assistant like' – not trying to mimic human emotions."

## A.5  Human Evaluation Instructions

Instructions for the intent-preserving task: "You will receive a dialog and a query, and need to label 1 if the dialog has the same intent as the query. If you are unsure, you can check to see whether both intents return the same Google search results.".

Table 4: Random Examples from Auto-Generated QRECC

Dialog:
User: who plays the lead role in wish upon a star
Assistant: The cast of Wish Upon a Star include Katherine Heigl, Danielle Panabaker, and Mary Page Keller.
User: who plays haley
Question: who plays haley in wish upon a star?
Answer: Danielle Harris

Dialog:
User: where is assam
Assistant: Assam is a state in India
User: what is the capital
Assistant: The capital of Assam is Dispur
User: who lead the first revolt against british in the state
Question: who lead the first revolt against british in assam?
Answer: Gomdhar Konwar

Dialog:
User: when did last man standing air
Assistant: Last Man Standing premiered on ABC on October 11, 2011.
User: what is the main character's name
Assistant: Tim Allen stars as Mike Baxter, the central character of the series.
User: what are the other character's names
Assistant: The other characters are Mike's wife and three daughters.
User: who plays cami harris
Question: who plays cami harris on last man standing?
Answer: Sarah Gilman

Dialog:
User: who were the confederates in the civil war
Assistant: The Confederates in the American Civil War were the Confederate States of America, or the South.
User: who fought them
Question: who fought the confederates in the american civil war?
Answer: The Union

Dialog:
User: when did adele webber first come on grey's anatomy
Assistant: The character of Adele Webber was first introduced in the episode ""Save Me"".
User: who plays her
Question: who plays adele webber on grey's anatomy?
Answer: Loretta Devine

Dialog:
User: what did the effigy mounds look like
Assistant: The effigy mounds are earthen mounds built in the shapes of animals, humans, and other objects.
User: when were they discovered
Assistant: The first effigy mounds were discovered in the 1700s but were not identified as such until the late 1800s.
User: when were most of them built
Questions: when were most of the effigy mounds built?
Answer: 350 - 1300 CE

Instructions for the natural vs. auto-generated task: "Can you guess which of the following dialogs is natural and which is auto-generated by a machine? There are 200 dialogs. 100 are synthetic, and 100 are natural. Enter 1 for Synthetic and 0 for Natural in the "synthetic?" field."

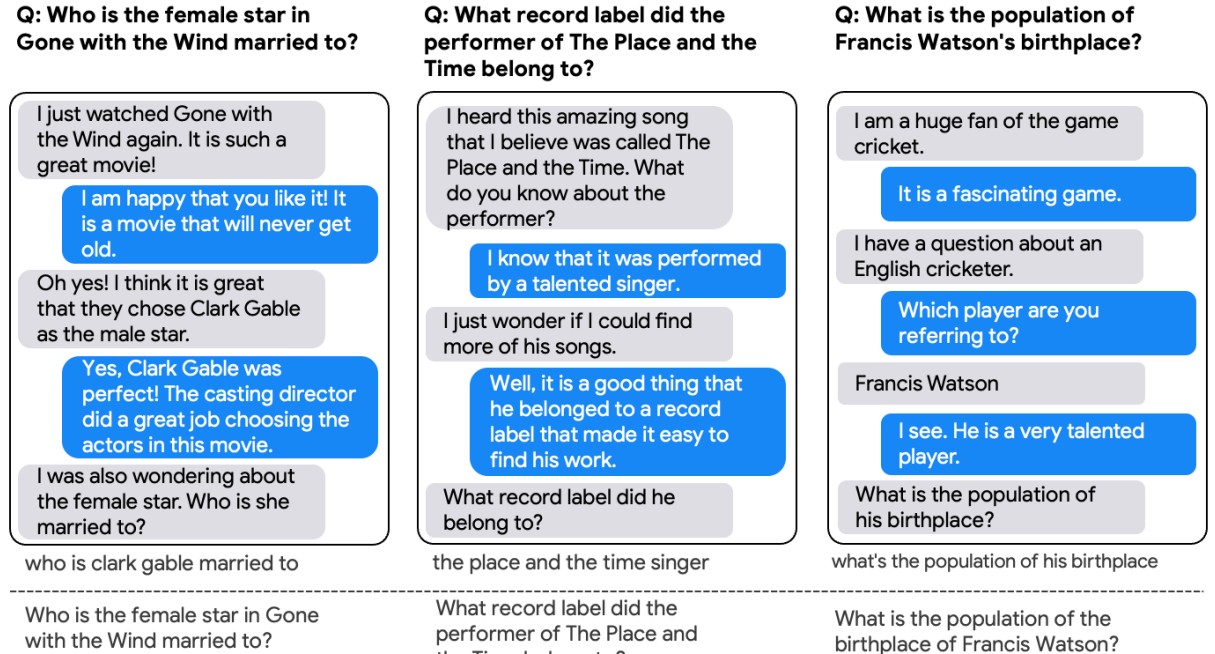

Figure 6: Examples from the **human-annotated** dialogues for Bamboogle. The model predictions above/below the line are of T5, before/after fine-tuning on MuSiQue dialogues.

Figure 7: Examples from the **auto-generated** dialogues for MuSiQue. The model predictions above/below the line are of T5, before/after fine-tuning on MuSiQue dialogues.

## A.6 Additional Examples

Table 9 shows examples of the ground-truth vs. the predicted queries from samples in the Bamboogle dialogs dataset.

## A.7 Parlai Models

We used available versions in Parlai (Miller et al., 2017) that can be accessed through an open API for a popular search engine. For the WizInt model

we used: zoo:sea/bart_fid_sqse/model. For Blende-Bot3 model we used zoo:bb3/bb3_3B/model -o gen/r2c2_bb3. The search engine is: `https://github.com/scy6500/ParlAI_SearchEngine`

Table 5: Random Examples from Auto-Generated MuSiQue

Dialog:
User: I am looking for a movie to watch tonight.
Assistant: I think I can help you find a good movie. What are your preferences?
User: I like watching action movies.
Assistant: What about The Bourne Deception? It is a very popular action movie.
User: Who plays the main character of this movie?
Question: Who plays the main character of The Bourne Deception?
Answer: Matt Damon

Dialog:
User: I just watched Charlie and the Chocolate Factory and I can't get it out of my mind!
Assistant: What an amazing movie! I love it!
User: I have an argument with my sister about the spoilt character.
Assistant: That's the thing - there are so many characters in this movie that are not just black and white. They are complex characters.
User: I know the spoilt character is a boy but can you tell me who played him?
Question: Who played the spoilt character in Charlie and the Chocolate Factory?
Answer: Julie Dawn Cole

Dialog:
User: I was watching the movie 'Dolores Claiborne' yesterday. It made me curious about the producer of the movie.
Assistant: It is the movie with Kathy Bates, right?
User: Yes, and I was wondering about the producer, do you know who his spouse is?
Question: Who is the spouse of the producer of Dolores Claiborne?
https://www.overleaf.com/project/644a2e0b82237321a4a407f1
Answer: Helen Mirren

Dialog:
User: I have this friend that is totally into football.
Assistant: I see.
User: He is a big fan of the Al Janoub Stadium.
Assistant: I am familiar with the Al Janoub Stadium. It is a very modern stadium, and its design is very unique.
User: I'm not surprised. I'm wondering what league includes the operating group of the stadium.
Question: What league includes the operating group of Al Janoub Stadium?
Answer: Qatar Stars League

Dialog:
User: I am really into books these days. I just finished reading 'The Final Testament of the Holy Bible' and can't get it out of my mind.
Assistant: I can see how this book would be interesting. A lot of people think about books like that.
User: It is a really good book. I think that this book was published by a great publisher.
Assistant: Yes, it is. It is one of the best in the UK.
User: Do you know who founded the publisher of this book?
Questions: Who founded the publisher of The Final Testament of the Holy Bible?
Answer: Larry Gagosian

Table 6: Examples from the generated QReCC data with different SBERT similarity scores between original Query ($Q$) and the reversed Query ($R_Q$). The higher the similarity, the more the dialog's intent is the same as the original query. We took only dialogs with SBERT similarity $\geq 0.999$.

SBERT Similarity: 0.999
Dialog:
User: who is the chairman of the joint chiefs of staff
Assistant: General Joseph Dunford is the current Chairman of the Joint Chiefs of Staff.
User: who does he advise
$Q$: who does the chairman of the joint chiefs of staff advise
$R_Q$: Who does the chairman of the joint chiefs of staff advise?

SBERT Similarity: 0.75
Dialog:
User: who designed magic the gathering
Assistant: Richard Garfield is the creator of the Magic: The Gathering collectible card game.
User: who originally published the game
Assistant: Wizards of the Coast, a subsidiary of Hasbro, Inc. is the original publisher of Magic: The Gathering
User: who is the current publisher
$Q$: who created magic the gathering?
$R_Q$: who is the current publisher of the game Magic: The Gathering?

SBERT Similarity: 0.5
Dialog:
User: what year did the Miami Hurricanes win the National Championship
Assistant: The Miami Hurricanes won the national championship in 2001.
User: was the team undefeated
Assistant: The Hurricanes finished the 2001 season 12–0.
User: who did they win against in the national championship game
$Q$: who did the Miaimi Hurricanes win against in 1983
$R_Q$: Who did the Miami Hurricanes play in the national championship game

SBERT Similarity: 0.25
Dialog:
User: who won the golden glove award at the 2018 fifa world cup
Assistant: Thibaut Courtois was named the winner of the Golden Glove at the 2018 FIFA World Cup.
User: what team did he play for?
$Q$: who won the golden glove at the 2018 fifa world cup?
$R_Q$: what team did thibaut courtois play for?

SBERT Similarity: 0
Dialog:
User: who inspired the book wonder by RJ Palacio
Assistant: The book Wonder by RJ Palacio was inspired by a childhood story of a boy with a severe facial difference.
User: what was the name of the boy
Assistant: The name of the boy was Julian.
User: what was his facial difference
Assistant: The boy had a craniofacial condition called Treacher Collins syndrome.
User: what is the condition
$Q$: what inspired the book Wonder
$R_Q$: what is Treacher Collins syndrome

Table 7: MuSiQue Auto-Generated Dialogues Prompts

Write a dialog between an automated assistant and a user, and the dialog should indirectly ask the initial question you received.

Question: When was the institute that owned The Collegian founded?
Dialog:
User: I have this homework that I need to submit in my history of the journalism course. Can you help me find out some of the details?
Assistant: Sure, I am here to help
User: I am working together with my friend Darren. We are looking into different newspapers, focusing on the powers that own them. I'm currently looking into The Collegian
Assistant: I can find out about The Collegian. It is actually owned by an educational institute.
User: When was the institute founded?

Question: What city is the person who broadened the doctrine of philosophy of language from?
Dialog:
User: I am conducting some research in the area of doctrine of philosophy of language
Assistant: I see. It is a fascinating sub-field of linguistics. It developed in quite an interesting process.
User: I know that it was broadened by some important philosopher
Assistant: Indeed, a famous german philosopher broadened it by focusing on language and its limits and ambiguities
User: What city was he from?

Question: Who employs the person who wrote the book Animal Liberation?
Dialog:
User: I just finished reading the book Animal Liberation. To be honest, I can't get the book out of my mind! Assistant: Indeed, some people find animal liberation extremely important. Others care a lot more about humans and do not view this topic as very important.
 User: I wish I could write such influential books one day. I am not sure if my company would even let me do that.
Assistant: Well, fortunately for the Australian writer of the book, he works for a company that allowed him to fulfill himself and publish the book
User: Which company was that

Table 8: QRECC NQ Auto-Generated Dialogues Prompts

Write a dialog between an automated assistant and a user, and the dialog should indirectly ask the initial question you received.

Question: Why was the great wall built?
Dialog:
User: where is the the great wall of china located
Assistant: The Great Wall of China is built across the historical northern borders of China.
User: how long is the wall
Assistant: The Great Wall is 21,196 km (13,171 mi).
User: why was the wall built

Question: Which U.S. states produce the most crued oil?
Dialog:
User: What kind of oil is in North Dakota
Assistant: The oil-rich Bakken shale formation has made North Dakota the second-largest crude-producing state behind only Texas.
User: Why
Assistant: North Dakota generated 852 million barrels of Bakken crude, The Bakken and the Three Forks formation are 94% of Dakota's current oil production, with about 1 million barrels a day.
User: In how many U.S states do we find the production
Assistant: Crude oil is produced in 32 U.S. states and in U.S. coastal waters.
User: Which ones produce the most

Question: Where is henry cavill from?
Dialog:
User: where is superman in the justice league movie
Assistant: In the Justice League Movie, Superman has been dead for two years and mankind is in mourning him.
User: who plays the character
Assistant: Henry Cavill plays Superman in the Justice League movie
User: where is the actor from

| Question | BlenderBot3 Predictions |
|---|---|
| What is the maximum airspeed (in km/h) of the third fastest bird? | Fastest bird |
| Who directed the highest grossing film? | Top grossing films |
| The husband of Lady Godiva was Earl of which Anglic kingdom? | Lady Godiva |
| Who founded the city where the founder of geometry lived? | Geometry |
| Who was the head of NASA during Apollo 11? | Apollo 11 |
| When was the longest bridge in the world opened? | Longest bridge |
| When was the anime studio that made Sword Art Online founded? | Sword Art Online |
| What is the capital of the country where yoga originated? | Yoga origin |
| Who is the father of the father of George Washington? | George Washington father |
| Who was the first king of the longest Chinese dynasty? | first king of the longest dynasty |

Table 9: Examples for the ground-truth queries from the Bamboogle dialogues with the BlenderBot3 queries. The BlenderBot3 seems to be trained more on finding the topic than asking concrete questions.