# OpenReview forum: "q2d: Turning Questions into Dialogs to Teach Models How to Search"
_EMNLP/2023/Conference — EMNLP 2023 Main_

### Official Review · Reviewer_exvd · 2023-08-04

**Soundness:** 4

**Excitement:**

4: Strong: This paper deepens the understanding of some phenomenon or lowers the barriers to an existing research direction.

**Paper Topic And Main Contributions:**

This paper turns a QA dataset into dialog conversation by prompting a PaLM. The mapping between dialog to query later on is exploited to train and evaluate query generation models. This query generation model is useful when combined with the search API, this is bridging the conversational dialog with the factual answer from the search API. Through evaluation, the author argues that the conversation produced by this approach to be natural, factual, and correct.

**Questions For The Authors:**

1) (page 5 line 367) Is it not really clear how the author produces a partially decomposed query. The paper describes making 2 query generation calls to a decomposed query. How does the author know that the query can be decomposed into 2 calls? In this study, are there no query that can be broken down into 3 or more calls?
2) (page 7 line 502) What does it mean by intent-preserving? The author cannot see the correlation between the term intent-preserving and full annotation instruction in the appendix. Maybe this can be clarified easily by explaining what is the intent-preserving in 1-2 short sentences.

**Reasons To Accept:**

The author proposes a way to approach hallucinations in conversational LLM. The proposed approach bridges the gap between LM and grounded knowledge by deconstructing dialog to search query automatically. This is interesting and can provide insight to the Dialog/NLP researcher. Overall, this paper is complete and gives a full analysis on every experiment step.

**Reasons To Reject:**

There is no strong reason to reject this paper, however there are some improvements that can be done to improve the clarity of the paper, it can be found below in the question for the authors section.

**Reproducibility:**

5: Could easily reproduce the results.

**Reviewer Confidence:**

5: Positive that my evaluation is correct. I read the paper very carefully and I am very familiar with related work.

---

> ### Author Rebuttal · Authors · 2023-08-24
>
> We thank you for engaging with us to improve our work. We appreciate the constructive comments and your acknowledgment  that the proposed approach bridges the gap between LM and grounded knowledge.
>
> ### Partially Decomposed Query
> We appreciate your keen observation on query decomposition. To clarify, our paper emphasizes a single technique for handling dependent or "multi-hop" queries in dialogues. For instance, a dialog about the US president followed by "How old is he?" could lead a model trained in 2020 to generate a query like "*How old is Donald Trump?*", and we produce a query asking how old is the current US president without assuming their identity. By focusing on this method, we aim to avoid incorporating outdated or incorrect information from the model. This specific choice is outlined in line 382: "*The fine-tuning technique described in this section uses the last approach to avoid making assumptions about the current president’s age or identity*.". We acknowledge that this could be made clearer in the text, and we will revise accordingly.
>
> ### Intent Preserving Query-Dialog Pairs
> Thank you for noticing that. To clarify, the term refers to ensuring that the dialog and the query share the same core intent. We'll specify the meaning of "intent-preserving" at its first mention at line 186: "*dialogs whose intent is different from the original query*". The full explanation is available in the annotators' guidelines, as noted in footnote 6 and line 858: "*Annotators should confirm that the dialog and query share the same intent, validated by comparable Google search results*."

---

### Official Review · Reviewer_5Dsm · 2023-08-05

**Typos Grammar Style And Presentation Improvements:** None that I saw.
**Soundness:** 3

**Excitement:**

4: Strong: This paper deepens the understanding of some phenomenon or lowers the barriers to an existing research direction.

**Paper Topic And Main Contributions:**

This paper focuses on the problem of generating dialogs from complex queries, with the goal of teaching conversational models how to search for information, e.g. on a user's behalf. There are a number of contributions made to the space in this paper, as follows:

1) a method for generating dialogs from complex queries
2) experiments showing that the generated dialogs are useful for training dialog models
3) an analysis of the generated dialogs comparing them with human written dialogs

**Questions For The Authors:**

Regarding table 2, I find it interesting that performance of the weaker baselines (first three rows) is generally lower on "B" vs. "M" - while for the stronger baselines (and the fine-tuned model proposed by the authors) that is not the case. It would be interesting to comment on this in the paper, in particular as the behavior of T5 itself changes after fine-tuning.

I find the example shown in figure 3 somewhat unconvincing. The final turn in the dialog, while perhaps not filtered because it did not meet the filter criteria (e.g. >80% overlap with the original query), nevertheless seems substantially similar to the original query in terms of its complexity (or number of intermediate searches required). It would be interesting to comment on how often this happens in practice and how to capture this particular aspect of filtering, which should make for more human-like conversations being generated (or at least kept) in the final datasets used for later experiments.

**Reasons To Accept:**

The paper introduces a useful new method for producing dialogs from queries, which is proven in experiments to help improve dialog model performance compared to a number of meaningful baselines. The dialogs generated are found to be substantially similar to human dialogs when evaluated by human annotators. The experimental results are generally fairly convincing and the analysis is relatively detailed. Finally, the authors promise as a contribution to make publicly available both the code and the datasets.

**Reasons To Reject:**

I have two reasons to argue against acceptance.

One is related to the overall clarity of the paper. I found the paper generally hard to read and sometimes information provided in one section did not align very well with information provided later. As an example, the abstract introduces a different set of contributions from those listed in section 1.

Second, while I think that the set of baselines included in experiments summarized in table 2 is reasonable, I am wondering whether the choice of adaptation only using T5 (whose performance out of those baselines is average) is the most convincing option. Applying the fine-tuning process to the other models used as baselines would give a clearer picture as to how resilient the approach is to fine-tuning different base models. There are other observations about table 2 results which are not discussed in the paper - I will include those as questions, below.

**Reproducibility:**

4: Could mostly reproduce the results, but there may be some variation because of sample variance or minor variations in their interpretation of the protocol or method.

**Reviewer Confidence:**

4: Quite sure. I tried to check the important points carefully. It's unlikely, though conceivable, that I missed something that should affect my ratings.

---

> ### Author Rebuttal · Authors · 2023-08-24
>
> We thank the reviewer for their constructive feedback. We are happy the reviewer acknowledged the quality of our experimental setup, the enhanced similarity to human dialogs, and our contribution of publicly available code and datasets.
>
> ### Abstract vs. Section 1 Information
> We appreciate your concern about the discrepancy between the abstract and Section 1. To clarify, the first and second points in the abstract about "query generation on the QReCC dataset" and "data generation in new domains like multi-hop MuSiQue and Bamboogle QA" correspond directly to the second point in Section 1. Similarly, the third point in the abstract about "thorough analysis" aligns with the third point in Section 1. We'll review the document to make it clearer and easier to read.
>
> ### Choice of T5 Model in Table 2
> Regarding the selection of T5 for fine-tuning, the key takeaway from Table 2 is that even a relatively smaller model like T5-3B can achieve superior results when utilizing our data generation pipeline, outperforming much larger models like PaLM 540B and Flan-U-PaLM 540B. We are confident that fine-tuning bigger models would likely produce better results. **We are also open to executing it if requested for this review**. However, we believe that a strong testament for our data generation method's value is showing that it enables *the smallest model to surpass the largest model* when used.
>
> ### Baseline Performance in Bamboogle vs. MuSiQue
> For context, Bamboogle's dialogs are sourced from human annotations, while MuSiQue's are auto-generated. We hypothesize that the top three models may not be as adept at handling auto-generated dialogs, hence their struggles. On the contrary, models like PaLM 540B, which generated the auto-test set, and its fine-tuned counterpart Flan-U-PaLM 540B, are naturally more aligned with generated dialog styles. T5, after training on a MuSiQue-styled dataset (ensuring query and dialog uniqueness), better adapts to this distribution.
>
> ### Figure 3 Example + Filter
> The semantic similarity score between the queries you pointed out in Figure 3 is 0.5034, as calculated by the SentenceTransformer model we used. Importantly, the latter query doesn't reference the movie "Gone with the Wind," which brings a different semantic nuance captured by the model. As stated in line 207, the filtered cases are ones that “include situations where no anaphora is required.”. For example, if the last turn was “BTW, about the female star in Gone with the wind. Any idea who is she married to?” This turn is still human-like, but does not require anaphora. It has a similarity of 0.8924, and therefore it will be filtered. The few-shot examples did not include such examples, and the overall percentage of these “non-anaphora requiring dialogs” did not surpass 5%.

---

### Official Review · Reviewer_GsVR · 2023-08-06

**Typos Grammar Style And Presentation Improvements:** NA
**Soundness:** 4

**Excitement:**

4: Strong: This paper deepens the understanding of some phenomenon or lowers the barriers to an existing research direction.

**Missing References:**

NA

**Paper Topic And Main Contributions:**

Given that manually-dataset-annotaion is usually time and resource consuming, this paper proposes an automatic data generation pipeline to generate information-seeking dialogs from questions.

To create a dialog, the pipeline starts from a search query and then generates an information-seeking dialog that corresponds to that query.

Extensive experiments demonstrate that (1) the method is beneficial for training query generation and dialog generation; (2) the generated dialogs are natural, factual and correct.

**Questions For The Authors:**

The dialogs are generated by foundation model, whether or not they may have limited assistance in improving the effectiveness of the original foundation model itself.

**Reasons To Accept:**

(1) Automatic data generation is a critical challenge, especially in the era of foundation model;
(2) It is essential to Enhance foundation models with the ability to independently search for relevant information to ground a given dialog response.
(3) The proposed method is effective and easy to implement.

**Reasons To Reject:**

NA

**Reproducibility:**

4: Could mostly reproduce the results, but there may be some variation because of sample variance or minor variations in their interpretation of the protocol or method.

**Reviewer Confidence:**

4: Quite sure. I tried to check the important points carefully. It's unlikely, though conceivable, that I missed something that should affect my ratings.

---

> ### Author Rebuttal · Authors · 2023-08-24
>
> We appreciate the reviewer's positive comments and acknowledgement of our method as an effective and easy to implement method for enhancing foundation models with the ability to search independently..
>
> ### Can Generated Data by a Foundation Model Improve the Original Foundation Model?
> Thank you for highlighting this point. In our experiments, we primarily focused on demonstrating that the data generated by larger foundation models (e.g., PaLM 540B) can substantially benefit smaller models (e.g., T5-3B). Specifically, we showed that fine-tuning these smaller models on our generated data allows them to achieve performance that surpasses the performance of the original, larger models in query-generation tasks. While we did not explicitly test the improvement of the original foundation model (e.g., PaLM) using the generated data, it is an intriguing direction for future work. We will clarify this in the revised manuscript.

---

### Meta-Review · Area_Chair_Gmuf · 2023-09-25

**Recommendation:** 5

**Metareview:**

Important data generation pipeline using LLMs. Intelligent use of search API to aid the generation of information-seeking dialog. It'd be valuable to community if the framework/pipeline/code is open-sourced for reproduction and further extension.

---

### Decision · Program_Chairs · 2023-10-07

**Decision:**

Accept-Main

**Comment:**

Important data generation pipeline using LLMs. Intelligent use of search API to aid the generation of information-seeking dialog. It'd be valuable to community if the framework/pipeline/code is open-sourced for reproduction and further extension.